# Inference Attacks Against Face Recognition Model without Classification Layers

## Abstract

Face recognition (FR) has been applied to nearly every aspect of daily life, but it is always accompanied by the underlying risk of leaking private information. At present, almost all attack models against FR rely heavily on the presence of a classification layer. However, in practice, the FR model can obtain complex feature embedding of the input via the model backbone, and then compare it with the target for inference, which does not explicitly involve the outputs of the classification layer adopting logit or other losses. In this work, we advocate a novel inference attack composed of two stages for practical FR models without a classification layer. The first stage is the membership inference attack. Specifically, We analyze the distances between the intermediate features and batch normalization (BN) parameters. The results indicate that this distance is a critical metric for membership inference. We thus design a simple but effective attack model that can determine whether a face image is from the training data set or not. The second stage is the model inversion attack, where sensitive private data is reconstructed using a pre-trained generative adversarial network (GAN) guided by the attack model in the first stage. To the best of our knowledge, the proposed attack model is the very first in the literature developed for FR models without a classification layer. We illustrate the application of the proposed attack model in the establishment of privacy-preserving FR techniques.

## 1 Introduction

Face recognition (FR) [5, 31, 18, 21, 13] technology has improved steadily over the past few years. It has been widely applied in a large number of personal or commercial scenarios for enhancing user experience. Recent studies [3, 27, 36] indicated that existing FR models can remember the information from the training data, making them vulnerable to some privacy attacks such as model extraction attacks [29], model inversion attacks [6], attribute inference attacks [7] (also known as property inference attacks) and membership inference attacks [26]. As a result, malicious attackers may be able to obtain users' private information through the use of certain attacks, which could cause significant damage. In order to facilitate the development of privacy-preserving FR methods, it is essential to evaluate the leakage of the private information quantitatively. This motivates this study which leads to the establishment of novel inference attacks to quantify such privacy leakage.

Inference attacks on machine learning algorithms can be roughly categorized into *membership inference attacks* and *model inversion attacks* [24]. We shall focus on both types of attacks against FR models. The goal of a membership inference attack is to infer whether a record is included in the training data set or not, which is usually formulated as a binary classification problem. Model inversion attacks attempt to recover the input of the target model. For example, some inversion attacks can recover the identity information of the training data from the FR model.

Submitted to 37th Conference on Neural Information Processing Systems (NeurIPS 2023). Do not distribute.

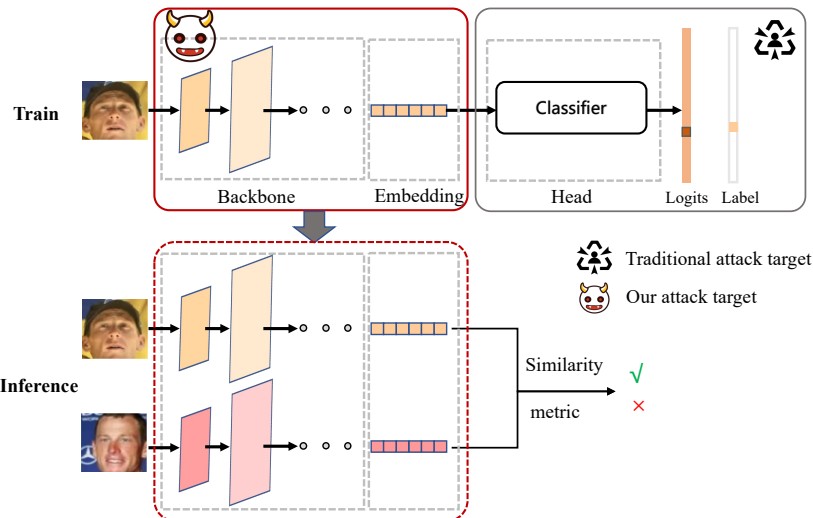

Figure 1: The schematic diagram of the training and inference process of the FR model. Our attack target is the backbone, while traditional attacks focus on the output of the classification layer which can be removed in the generalized FR process.

At present, both types of attacks against FR usually rely on the presence of a classification layer. In other words, the performance of the attack and the corresponding defense algorithms [33] is heavily dependent on utilizing the output from logits in the classification layer or the labels themselves [4]. Nevertheless, FR has completely different training and inference paradigms, as shown in Fig.1. Existing face recognition techniques do include a classifier after the feature extraction network (backbone) in training, but the classifier is not used during inference. Instead, they extract feature embeddings from the image pair using the trained backbone, then calculate their similarity based on certain metrics to determine whether the two images are from the same person to achieve FR. Therefore, in a realistic setup, the attacker can only acquire the feature embedding of the query image and has no access to the discarded classification layer. The aforementioned difference in the paradigms of training and inference for FR brings significant challenges to the current attack methods, because there is evidence [8] indicating that compared with logits and losses, the feature embedding, as a more general representation, contains *less* information about the training data. Moreover, the training stage of an FR system is essentially a closed-set classification problem, while the inference phase becomes a more complicated open-set problem. In summary, performing inference attacks against FR without exploiting the classification layer is non-trivial but practically important.

In this paper, we shall propose a novel two-stage attack algorithm, and its diagram is shown in Fig.2. The first stage is the membership inference attack. Compared to previous attack methods that focus on the classification layer of the target model, we explore the inherent relationship between the backbone output of the target model and the training data set. Specifically, we first analyze the distribution of the distances between the intermediate features of the member/non-member samples and statistics stored in the Batch Normalization layer in the trained backbone, and found it critical for membership inference. We then design an attack model to determine whether a sample belongs to the training data set or not, through utilizing this distance distribution. We conduct experiments under different levels of prior knowledge including partial access to training data set and no access to training data set. In both cases, the proposed membership attack is efficient and can provide state-of-the-art performance.

The second stage is the model inversion attack. Previous model inversion attacks against FR such as those from [28, 1, 17] are heavily dependent on the classifier of the target model. They often guide the optimization of the attack network using the logit of the classifier's output or some well-designed loss functions based on the logit. For the FR models in consideration, none of these approaches are applicable. We thus put forward a novel inversion attack approach, where our attack model in the first stage guides the synthesis network, StyleGAN, to optimize the latent code such that the synthesized face becomes close to a member sample in the training data set as much as possible. Specifically, we adopt the PPA [28] paradigm, preferring to sample a random batch of samples, filtering out these samples that are more likely to be considered members of the training data set by the first-stage attack

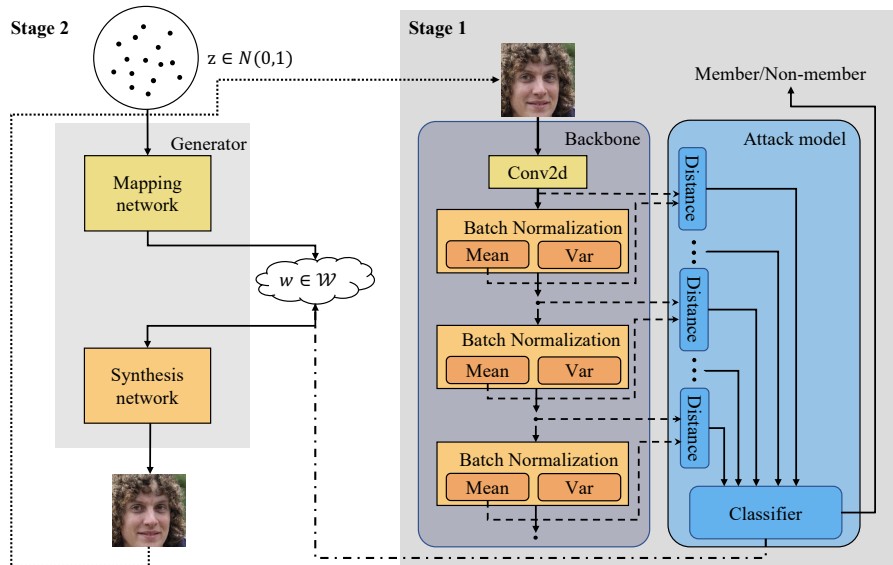

Figure 2: The diagram of the proposed two-stage inference attack against FR models without classification layers. The first stage is the membership inference attack. The attack model utilizes the parameters from the Batch Normalization layers to determine whether a sample belongs to the training data set or not. The second stage is the model inversion attack. StyleGAN is adopted to synthesize images and optimize the results from the output of the first-stage attack.

model, and then further optimizing the remaining samples to make them better fit the membership distribution. Experiments demonstrate that the proposed attack algorithm against FR can recover some sensitive private information in the training data set.

Our contributions can be summarized as follows:

- We extend existing inference attacks against FR models by relaxing the assumption of the existence of a classification layer in the inference process. The proposed attack method can thus make the defense techniques in literature targeting the FR classification layer subject to the risk of private information leakage again.

- We analyze the distance between the intermediate features of member/non-member samples of the training data set and the parameters of the BN layers. We design a simple and effective attack model accordingly, which is applicable to the trained FR models without classification layers. Experiments demonstrate that our method outperforms the state-of-the-art methods with the same prior knowledge.

- We propose a model inversion attack algorithm that does not require the classification layer. It makes the synthesized samples better fit the membership distribution, and its effectiveness is verified by our experiments.

## 2 Related Work

### 2.1 Membership Inference Attack

Member inference attacks aim to speculate whether a record has been involved in the training of the target model. In recent years, there have been significant efforts devoted to membership inference attacks and defenses. Specifically, [26] first proposed the membership inference attack, where the attacker achieves membership inference through multiple attack models. [25] relaxed the assumptions in membership inference attacks; other works [4, 20] proposed membership inference attacks for scenarios where only labels are available. In addition, [2] analyzed the membership inference attack from a causal perspective. Meanwhile, some works focused on defenses against membership inference attacks, and the available algorithms can be broadly categorized into two types: overfitting

reduction [30, 26, 25] and perturbation model prediction [23, 15, 33]. On the contrary, the attack method presented in this paper does not require the use of labels and model predictions. It not only achieves good attack performance, and more importantly, it renders the existing defense methods that are based on the presence of the classification layer vulnerable again.

Recently, attack techniques [19, 8] with no demand of the classification layers attracted great attention, which were developed mainly for pedestrian re-identification (Re-ID). FR is a more challenging task compared with pedestrian re-identification in the sense that the face data set contains only face-related information but has more identities and samples than the Re-ID data set. Moreover, [19] requires acquiring multiple samples of the same class, while [8] needs to obtain partial training data. These conditions are difficult to fulfill in realistic scenarios. The developed method in this paper outperforms the existing membership inference attack algorithms in terms of improved success rate under the same amount of available information.

### 2.2 Model Inversion Attack

The model inversion attack is a particular type of privacy attacks on machine learning models, which aims at reconstructing the data used in training. Many recent model inversion techniques [17, 28, 1, 10] have been implemented with the help of the generative power of GAN. [17] formulated model inversion as the problem of finding the maximum a posteriori (MAP) estimate given a target label, and limited the search space to the generative space of GAN. [28] proposed a more robust cross-domain attack with a pre-trained GAN, which can generate realistic training samples independent of the pre-trained data. MIRROR [1], on the other hand, explored a new space $\mathcal{P}$ over StyleGAN, and regularize latent vectors in $\mathcal{P}$ to obtain good performance. [10] cast the latent space search as a Markov decision process (MDP) and solved it using reinforcement learning.

Meanwhile, there are some works [33, 32, 22] focusing on defenses against model inversion attacks. [33] proposed a unified purification framework to resist both model inversion and membership inference attacks by reducing the dispersion of confidence score vectors. DuetFace [22] utilized a framework that converts RGB images into their frequency-domain representations and makes it difficult for attackers to recover the training data by splitting them. PPFR-FD [32] investigated the impact of visualization components on recognition networks based on the privacy-accuracy trade-off analysis and finally proposed an image masking method, which can effectively remove the visualization part of the images without affecting the FR accuracy evidently.

## 3 Method

In this section, we present the proposed attack method in detail, whose diagram is shown in Fig.2. We first briefly review the existing training and inference paradigms of FR in Section 3.2. Next, we introduce the difference in the distance distribution of member/non-member compared to BN layer parameters in Section 3.3, which will guide the member inference attack. In Section 3.3, we will formally introduce the first stage of the inference attack, i.e. membership inference attack algorithm. Furthermore, we will introduce our second stage - the model inversion attack method without classification layers in Section 3.4.

### 3.1 Background

Commonly adopted training and inference paradigms of FR are shown in Fig.1. Given a face data set $\mathcal{D}$, an image sampled from it is denoted as $x \in \mathbb{R}^{c \times h \times w}$, where $c$, $h$ and $w$ represent the number of channels, height and width of the sample. Its identity label can be represented as a one-hot vector $y \in \mathbb{R}^{n \times 1}$, where $n$ is the number of classes. In the training phase, a backbone network $B$ with model parameters collected in $\theta_b$ takes an input sample and extracts the corresponding high-dimensional feature embedding $v = B(x) \in \mathbb{R}^{f \times 1}$, where $f$ is the dimensionalities of the embedding. The backbone network $B$ employs a classification layer $C$, whose parameters are collected in $\theta_c$, to predict the class $\hat{y} = C(B(x))$ for the obtained embedding $v$. The cross-entropy, denoted as $\mathcal{L}_{CE}(\hat{y}, y)$, is usually used as the loss function in training, and the training process can be formulated as the following minimization problem with respect to $\theta_b$ and $\theta_c$:

$$\min_{\theta_b, \theta_c} \mathbb{E}_{(x,y) \sim \mathbb{D}} \mathcal{L}_{CE}[C(B(x)), y]. \tag{1}$$

During inference, the goal is to decide whether the two face images come from the same person or not. The trained backbone $B$ is utilized to extract the embeddings of the two face images without utilizing the classification layer anymore. The similarity of the obtained two embeddings is evaluated based on certain metrics. If the computed similarity is higher than a well-designed threshold, they are considered to be from the same person.

## 3.2 Distance Distribution Comparison

Current studies [19, 8] on the distributions of feature embeddings of samples from the training data set (i.e., member samples) and other samples (i.e., non-member samples) were carried out. In particular, [19] requires a large number of inter-class samples to calculate the class centers, while ASSD [8] demands access to some training samples as reference samples to calculate their Euclidean distances. These assumptions may be too strict and are not always available to attackers. Thus, we shall look for information on the training samples stored in the trained backbone network $B$ itself.

Specifically, we consider replacing the reference samples required in [19, 8] with the statistics stored in the Batch Normalization (BN) [14] layer of the trained backbone network. A BN layer normalizes the feature maps during the training process to mitigate covariate shifts [14], and it implicitly captures the channel-wise means and variances [35]. To gain more insights, we compare the distances between the intermediate features of the member/non-member samples and the corresponding "running mean" and "running variance (var)" in several BN layers, and the results are shown in Appendix.

There is a clear boundary separating the distance distribution of member and non-member samples in some BN layers. This indicates that the BN layer parameters may be utilized for membership inference. This observation is fundamentally different from the findings in [19, 8], as they completely ignored the information stored in the trained model itself on the training data set. The performance of the techniques from [19, 8] heavily depends on the number of samples used. If the available samples are insufficient, they will not perform as well as expected. Our observation, on the other hand, relax these constraints on the following membership inference attack.

## 3.3 Inference Membership Attack

The proposed membership inference attack algorithm is designed based on the observation presented in the previous subsection. It is also the first step of our attack model, as shown as Stage 1 in Fig. 2. Specifically, we consider the "running mean" parameters in certain BN layers, denoted by $u_i \in R^{b_i \times 1}$, where $b_i$ represents the number of channels in the $i$th BN layer, and thus we obtain a set of vectors $u = \{u_1, u_2, \cdots, u_n\}$, where $n$ represents the number of BN layers selected. It is worthwhile to point out that since the number of channels in the BN layers can be different, the vectors $u_i$ in $u$ do not necessarily have the same dimensionality.

Given the input image $x$, we extract the intermediate features $v_i \in R^{c_i \times h_i \times w_i}$ before a particular BN layer, and then we obtain $\overline{v}_i \in R^{c_i \times 1}$ by normalizing along both the height and width dimensions following the BN operation. We then compute the Euclidean distance between the extracted and normalized feature $\overline{v}_i$, and the reference $u_i$ using

$$d_i = \frac{1}{b_i} ||\overline{v}_i - u_i||_2^2. \tag{2}$$

We transmit the distance vector $d = \{d_1, d_2, \cdots, d_n\}$ into the classification network $\mathcal{A}$. Due to the good distinguishability of the features we extract, we do not need to design a complex network for membership inference. Here, we only need to use a fully connected layer and a sigmoid function to compose our attack model, which predicts the probability that the sample is a member. If it is from the training data set, then the attack model should output 1, otherwise we expect it to output 0.

More empirical experiments show that the use of both the original face image and its horizontally flipped version is capable of enhancing the performance. Therefore, in the implemented membership inference scheme, we first flip horizontally an image $x$ to obtain $x'$, and then extract $\overline{v}_i'$ by following the same procedure used to find $\overline{v}_i$. The distance $d_i$ is now computed as

$$d_i = \frac{1}{b_i} ||\frac{\overline{v}_i + \overline{v}_i'}{2} - u_i||_2^2. \tag{3}$$

The classification network $\mathcal{A}$ is trained through solving

$$\min_{\theta_{\mathcal{A}}} \mathcal{L}_{CE}[\mathcal{A}(d), s] \tag{4}$$

where $\theta_A$ denotes the parameters of the classifier $\mathcal{A}$ in our attack model, and $s$ is the binary label (0 or 1). The training procedure of the membership inference attack is shown in **Algorithm 1**.

---

**Algorithm 1:** The training procedure of the membership inference attack (Stage 1)

**Input:** Member of training data $x_m$ and non-member data $x_n$, number of training iterations $M$.
**Output:** The optimal parameters of the attack model $\theta_{best}$.

1 Extract the parameters of Batch Normalization layers in the target FR model, denoted as $u$.
2 Set $\theta_{best} = \theta_0$ and $Acc_{best} = 0$, where $\theta_0$ denotes the initial parameters of the attack model $\mathcal{A}$ and $Acc_{best}$ records the best performance on the test set.
3 **for** $i = 0, 1, 2, \cdots, M - 1$ **do**
4      Horizontally flip the member data $x_m$ and non-member data $x_n$, and obtain $x'_m$ and $x'_n$, respectively.
5      Feed the target model with $x_m$, $x'_m$, $x_n$ and $x'_n$, and extract the corresponding intermediate features $v_m$, $v'_m$, $v_n$ and $v'_n$.
6      Compute the distance using Eq.3 and obtain the distance vectors $d_m$ and $d_n$.
7      Update the classifier parameters using the following equation:
8

$$\theta_{i+1} \leftarrow \theta_i - \alpha \nabla_{\theta_i} \left( \mathcal{L}_{CE}[\mathcal{A}(d_m), 1] + \mathcal{L}_{CE}[\mathcal{A}(d_n), 0] \right)$$

     where $\alpha$ denotes the optimization step.
9      Evaluate the model on the test set and obtain the accuracy $Acc_{i+1}$, and update $\theta_{best} = \theta_{i+1}, Acc_{best} = Acc_{i+1}$ if $Acc_{i+1} > Acc_{best}$.

---

## 3.4 Model Inversion Attack

Existing model inversion attacks require that target labels are given in order to improve the prediction score of the target identity. In this work, we consider a model inversion attack on the FR models without the classification layer. As such, our goal is not to reconstruct the original image when the target identity is known. Instead, we focus on recovering the identity of the training data set using the trained backbone model as much as possible.

The model inversion attack is Stage 2 of the proposed attack model in Fig. 2. The procedure of model inversion attack is shown in **Algorithm 2**. In particular, we utilize the powerful generative model StyleGAN [16] to help restore the identity information of the training data. We first collect samples from a normal distribution $z \in \mathcal{N}(0, 1)$, and these samples are fed into StyleGAN's mapping network to obtain a more disentangled subspace $\mathcal{W}$. All our subsequent optimizations will be performed in $\mathcal{W}$ space.

[28, 1] have pointed out that the selection of the initial latent vectors to optimize has a strong impact on the effectiveness of model inversion attacks and its importance should not be underestimated. As the classification layer is absent in our case, it is not feasible to obtain the confidence of the target label, and we have to utilize the attack model in Stage 1 to select samples that are closer to the distribution of the training set. For this purpose, we feed all the initial latent vectors into the StyleGAN and produce a set of initial face images. After proper resizing, these images with their horizontally flipped versions are processed following Stage 1 and we can get the prediction of the developed attack model $\mathcal{A}$. Those images with high membership classification scores are retained as the 'final' set of initial vectors.

Next, we optimize the latent vectors as follows. In particular, we perform some operations on the images synthesized using the latent vectors, and propagate the transformed images through the developed membership inference attack model in Stage 1. The membership classification network predicts the probability that the generated images belong to the training set. The latent vectors will then be iteratively updated to further increase the membership prediction probability. We desire to enhance the robustness through data augmentations and finally find a good set of latent vectors.

**Algorithm 2:** The procedure of the model inversion attack (Stage 2)

---

**Input:** Initial $N$ points sampled from a normal distribution $z \in \mathcal{N}(0,1)$, number of iteration $M$.

**Output:** The generated candidate set $q$ (the size of $q$ is Int $(0.1N)$, Int $(0.1N)$ is the integer part of $0.1N$), and the realistic training image set $r$.

1 Generate $N$ images from initial sample points $z$ by the generator.

2 Extract the parameters of Batch Normalization layers in the target FR model, denoted as $u$.

3 Feed the target model with the generated images and their horizontally flipped versions, and compute the distance using Eq.3, and then predict the probability of membership. (We simplify this step as MI, i.e., the membership inference.)

4 Select Top-$n$ ($n$= Int$(0.1N)$) sampling points in descending probability order as the candidates to optimize their intermediate representations $w_0^j (j = 1, 2, \cdots, \text{Int}(0.1N))$. The relationship between $z$ and $w$ is shown in Fig.2.

5 Set $q = \{\}$ and append optimized candidates to it as follows:

6 **for** $j = 1, 2, \cdots, Int(0.1N)$ **do**

7     Generate the image $x_0^j$ from the selected sample point $w_0^j$ by the generator.

$$x_0^j = \mathcal{G}(w_0^j)$$

8     **for** $i = 0, 1, 2, \cdots, M-1$ **do**

9         Perform some data augmentation operations on the image $x_i^j$ and get the counterparts $x_{i1}^j, x_{i2}^j, \cdots, x_{im}^j$, where $m$ denotes the number of the data augmentation types.

10         Make membership inference and compute the average probability:

$$p_i^j = \frac{1}{m+1}[\text{MI}(x_i^j) + \sum_{k=1}^{m}(\text{MI}(x_{ik}^j))]$$

11         Optimize the sample point $w_i^j$ by minimizing the loss function:

$$w_{i+1}^j \leftarrow w_i^j - \alpha_w \nabla_w \mathcal{L}_{CE}[p_i^j, 1]$$

        where $\alpha_w$ denotes the optimization step size.

12         Generate the updated images: $x_{i+1}^j = \mathcal{G}(w_{i+1}^j)$

13     Append the optimized candidate to the set $q = q \cup \{x_M^j\}$

14 For each candidate $x_M^j \in q$, search the similar image in the training data based on the cosine similarity, and obtain the image pairs.

15 Select Top-$n$ ($n = \text{Int}(0.01N)$) image pairs in descending similaritiy order and obtain the realistic training image set $r$.

---

## 4 Experiment

In this section, we will describe our experimental setup and results in detail. First, for the membership inference attack of Stage 1, we perform two different settings based on different prior information: the partial training data $\mathcal{D}_p$ and the auxiliary data $\mathcal{D}_s$. For the model inversion attack of Stage 2, we only use the auxiliary data $\mathcal{D}_s$.

### 4.1 Dataset and Target Model

We use two datasets for inference attack, CASIA-WebFace [34] and MS1M-ArcFace [5]. The CASIA-WebFace dataset is collected in a semi-automatic way from the Internet, and is usually used for face verification and face identification tasks. The dataset contains 494,414 face images of 10,575 real identities. MS1M-ArcFace is obtained by cleansing on MS1M [9], and it contains 5.8M images from 85k different celebs.

We use IR-SE-50 as the backbone, which combines an improved version of the vanilla ResNet-50 [11] with SENet [12], and use ArcFace [5] as the margin-based measurements (head). The target model is trained for 50 epochs with an initial learning rate of 0.1 and step scheduling at 10, 20, 30

Table 1: The attack success rate of the membership inference attack in the case 1. We also consider the case where the target models are trained with randomly flipped images, denoted as 'FR(flip)'.

| $\mathcal{D}_p$ | Proportion | | | | | |
|---|---|---|---|---|---|---|
| | 1% | 5% | 10% | 1%+FR(flip) | 5%+FR(flip) | 10%+FR(flip) |
| ASSD [8] | 57.20 | 57.32 | 65.42 | 55.31 | 56.28 | 60.45 |
| $\mathcal{A}_{mean}$ | 77.94 | 78.03 | 79.75 | 73.83 | 74.84 | 78.27 |
| $\mathcal{A}_{mean\&var}$ | 77.22 | 76.52 | 78.58 | 68.24 | 69.34 | 78.18 |
| $\mathcal{A}_{mean\&flip}$ | **91.94** | **91.74** | **91.94** | **97.37** | **97.46** | **97.42** |

Table 2: The attack success rate of the membership inference attack in the case 2.

| $\mathcal{D}_s$ | Target Model(Backbone+Head) | | |
|---|---|---|---|
| | IR-SE-50+ArcFace | IR-SE-50+CosFace | IR-SE-101+ArcFace |
| $\mathcal{A}_{mean}$ | 63.37 | 62.56 | 71.19 |
| $\mathcal{A}_{mean\&var}$ | 64.43 | 62.25 | 71.29 |
| $\mathcal{A}_{mean\&flip}$ | **87.30** | **82.93** | **85.04** |

and 40 epochs, using the SGD optimizer with a momentum of 0.9, weight decay of 0.0001. Previous researches [25, 2] have proved 'if a model is overfitted, then it is vulnerable to membership inference attack.' In this experiment, to avoid overfitting, we choose the FR model that has the best test accuracy in all epochs during training as the target model.

## 4.2 Membership Inference Attack

As above mentioned, we give two different cases. Case 1: access to part of the training dataset. Case 2: access to the auxiliary dataset. In the case 1, we choose CASIA-WebFace as the training data set. We use different proportions of training data for training the attack model, which uses ASSD [8] as the baseline. In the case 2, MS1M-ArcFace is used as the training data set. We train a shadow model to mimic the behavior of the target model [25]. Specifically, we first split the dataset $\mathcal{D}_s$ by half into $\mathcal{D}_{shadow}$ and $\mathcal{D}_{target}$. Then we split $\mathcal{D}_{shadow}$ by half into $\mathcal{D}_{shadow}^{member}$ and $\mathcal{D}_{shadow}^{non-member}$. $\mathcal{D}_{target}$ is used for the attack evaluation, it is also split into $\mathcal{D}_{target}^{member}$ and $\mathcal{D}_{target}^{non-member}$. All the $\mathcal{D}^{member}$ serve as the members of the (shadow or target) model's training data, while the other serves as the non-member data. For evaluation, we sample 60,000 images from members and non-members separately in both cases and use the attack success rate (ASR) as the evaluation metric.

Furthermore, we perform experiments with different settings for ablation study. First, we conduct an experiment case where the images are not flipped and the "mean distance" is fed to the attack model, denoted as $\mathcal{A}_{mean}$. And then we consider replacing the "mean distance" with "mean and variance distances", which both are input to the attack model, denoted as $\mathcal{A}_{mean\&var}$. Finally, we use "mean distances" of the normal image and the horizontally flipped one as fusion features to input the attack model, denoted as $\mathcal{A}_{mean\&flip}$. Specially in case 2, we use different combinations of the backbones and heads as the target models to validate the generalization of our methods. All results of the membership inference attack are shown in Tab.1 and Tab.2 As Tab.1 shows, in case 1, the performance of our attack significantly outperforms the baseline. We believe this is due to the fact that our selected BN-based features characterize the membership of the training set better than the reference sample selected in [8], despite our network design is more lightweight than the latter. And we are also able to achieve relatively good results in the $\mathcal{D}_s$ experiments in case 2, which validates the effectiveness of our method.

## 4.3 Model Inversion Attack

To our knowledge, this is the first time that the model inversion attack is launched against an FR model without the classification layer. Previous metrics always judge the accuracy of the generated images given a target label, which are obviously not applicable in this scenario. Therefore, we use a new metric compatible with our proposed scenario. To be specific, we will use the target model

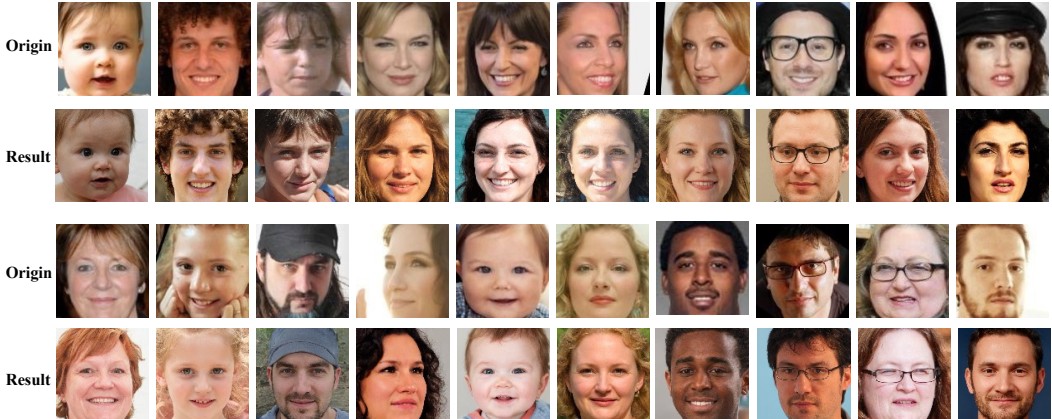

Figure 3: Some results of the model inversion attack in case 2. The first and third rows show the original training images, while the second and fourth rows show the synthesis images generated by our proposed algorithm without using the classification layer.

to extract the embedding of the final generated results, and obtain the most likely class index by means of the classification layer trained for FR. Then we search the most similar image pairs as mentioned in **Algorithm 2**. Some attack results are provided in Fig.3. The first and third rows show the original training images, while the second and fourth rows show the synthesis images generated by our proposed algorithm without using the classification layer. It can be seen that the reconstructed image is very similar to the original image (more results can be found in the appendix). This also confirms the effectiveness of our algorithm to some extent.

## 5 Limitation

Although we propose an inference attack algorithm against face recognition models without classification layers, there are actually some inherent assumptions that need to be considered. First, our membership inference attack is implemented based on the backbone's internal BN layer parameters, which means that our algorithm currently can only be applied to white-box attacks in such a scenario. Therefore, it is a future challenge to explore further attacks in a completely black-box scenario.

In addition, since we are not able to operate a delicate control on the attack, we cannot guarantee the expected results given the target class. And the identity characteristics of our final synthesis images still need to be improved, which is another major challenge in this scenario due to the fact that we cannot optimize the latent vector for a specific target label.

## 6 Conclusion

In this paper, we propose a new scenario where the inference attack against face recognition models can be implemented without the classification layers. This is a more realistic and more challenging attack, where the adversary cannot obtain information about the classification layer on the stage of training, and all the defenses based on the classification layer will be ineffective in this scenario. Considering the internal parameters of the model, we theoretically analyze the distance distributions between the member/non-member and the BN layer parameters, and accordingly design a simple and efficient attack model that is compatible with this novel scenario. Experiments demonstrate that our method outperforms state-of-the-art similar works under the same prior. Further, we propose a model inversion attack algorithm in this scenario. We utilize the classifier in our attack model to free model inversion attacks from dependence on the classification layer, and make the generated samples closer to the membership distribution. The final experimental results prove that our proposed method is able to recover the identities of some training members. We hope that this scenario and the algorithm we proposed will encourage more researchers to focus on face recognition attacks in real scenarios, and to attach further importance to privacy security protection.

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
