# 1 Comparsion of Feature Distances for the Member and Non-member

In this section, we give more details of experiments about the distance distribution comparison. We use a widespread network IR-SE-50 [2, 3] as the backbone, which consists of an input layer, an output layer and 4 sub-blocks. We extract the parameters "running mean" and "running var" in the last BatchNorm2d layer of each sub-block. In the backbone, the input and output layers include a BatchNorm2d layer separately. Besides, the output layer also has a BatchNorm1d layer. That is to say, we can obtain the parameters totally from 6 BatchNorm2d layers and 1 BatchNorm1d layer, and all the parameters are 1-dimensional vectors, denoted as $u \in R^{1 \times c}$. We represent the "running mean" and "running var" as $u_{mean}$ and $u_{var}$, respectively.

When computing the distance between the feature mean (or the feature) and "running mean", we average the intermediate feature $v_{2d} \in R^{1 \times c \times h \times w}$ (the feature before a BatchNorm2d layer) for the dimensions $h \times w$ and obtain the feature mean $\overline{v}_{2d} \in R^{1 \times c}$. Then we can compute the distance between $\overline{v}_{2d}$ and $u_{mean}$ using Eq.2 in the paper. And the intermediate features before a BatchNorm1d layer, denoted as $v_{1d} \in R^{1 \times c}$, have the same size as that of $u_{mean}$, so we can directly obtain the distance using Eq.2. And 7 distances are shown in Fig.1 .

Note that we also compute the distance between the feature *variance* and "running var" $u_{var}$. Specifically, we calculate the variance of $v_{2d} \in R^{1 \times c \times h \times w}$ for the dimensions $h \times w$ and obtain the feature *variance* $\hat{v}_{2d} \in R^{1 \times c}$, and then compute the distance as above. However, it makes no sense that we directly compute the distance between the variance of the feature before the BatchNorm1d layer and the corresponding $u_{var} \in R^{1 \times c}$. Since $u_{var}$ in the BatchNorm1d layer is obtained by calculating the variance of a batch of features along only the batch dimension $B$. Despite the size of the feature $v_{1d} \in R^{1 \times c}$ being the same as that of $u_{var}$, it does not provide any physical meaning if we perform mathematical operations on them directly. Therefore, we abandon the distance about the intermediate feature $v_{1d}$ and "running var", and obtain 6 distances as shown in Fig.2 .

# 2 Membership Inference Attack

In the paper, we use CASIA-WebFace [4] as the training data set in the case 1. When training and testing the attack model, we sample face images from another dataset as the non-members. Here we give a more challenging case. We split the one dataset $\mathcal{D}$ (CASIA-WebFace) by half into $\mathcal{D}^{member}$ and $\mathcal{D}^{non-member}$. And then we split $\mathcal{D}^{member}$ into $\mathcal{D}^{member}_{train}$ and $\mathcal{D}^{member}_{test}$. $\mathcal{D}^{member}_{train}$ is used to train the attack model and $\mathcal{D}^{member}_{test}$ for the test. We perform the same operation on $\mathcal{D}^{non-member}$ and obtain $\mathcal{D}^{non-member}_{train}, \mathcal{D}^{non-member}_{test}$. We also use different proportions of the dataset for training the attack model and the number of testing images is 60,000. Furthermore, we perform extended experiments with additional settings for ablation study. Besides the settings mentioned in the paper, we replace "mean distance" with "variance distance", in which case the attack model takes only "variance distance" as the input and the images are not flipped, denoted as $\mathcal{A}_{var}$. We also consider the case where both "mean distance" and "variance distance" are the input and the images are flipped, denoted as $\mathcal{A}_{mean\&var\&flip}$. We show the results in Tab.1. We find the attack success rate of the $\mathcal{A}_{var}$ case is lower than that of the $\mathcal{A}_{mean}$ case. In the cases without 'FR(flip)', the attack success rates of $\mathcal{A}_{mean\&var\&flip}$ and $\mathcal{A}_{mean\&flip}$ are similar. But in the cases with 'FR(flip)', the attack success rate of $\mathcal{A}_{mean\&var\&flip}$ is lower than that of $\mathcal{A}_{mean\&flip}$. From the above results, it can be seen that the introduction of $\mathcal{A}_{var}$ did not bring more performance gains. Therefore, in the experiments of the paper, we did not consider the case using $\mathcal{A}_{var}$.

# 3 Model Inversion Attack

We provide more results of our model inversion attack algorithm as shown in Fig3. We try to reconstruct identities as many as possible, but find the number of reconstructed identities is less than the case with a classification layer. We believe that the main reason for this result is the lack of classification layers, which includes much privacy information. However, to our knowledge, this is the first time that the model inversion attack is launched against an FR model without the classification layer.

**Distance between features and 'running mean'**

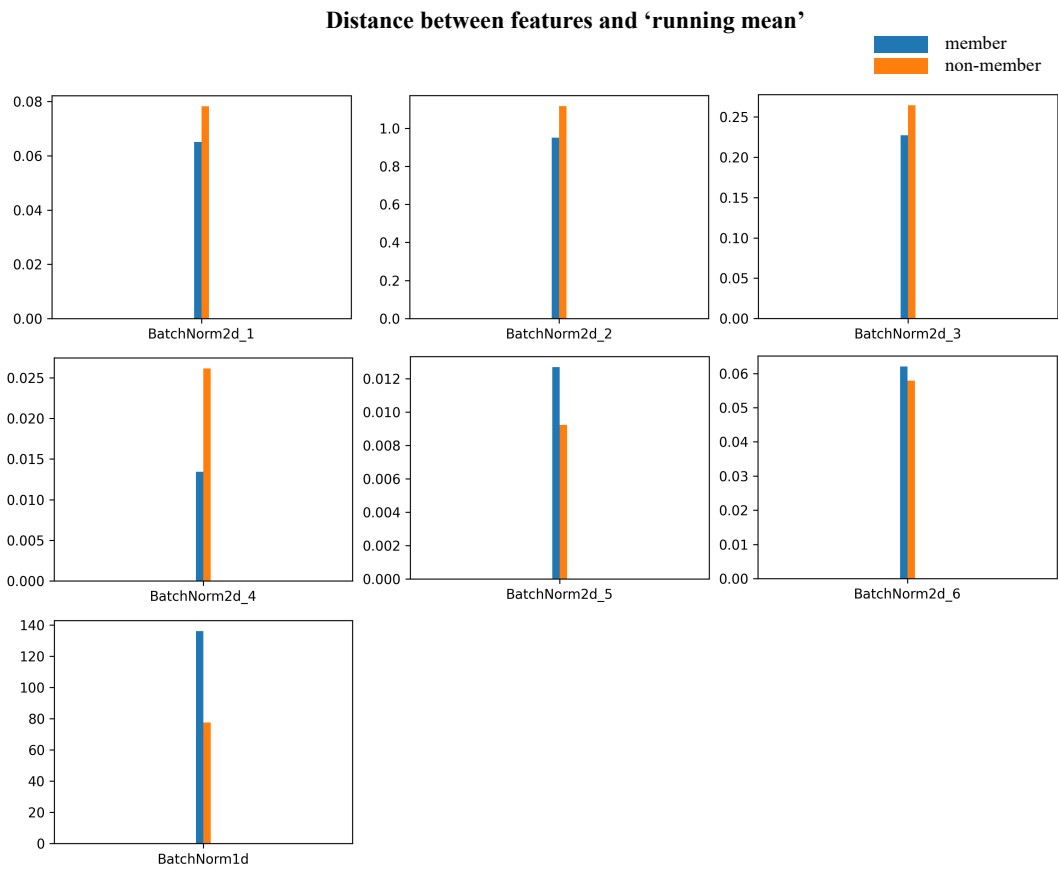

Figure 1: Visualization of distances between the intermediate features and "running mean" for the Member and Non-member.

**Distance between features and 'running var'**

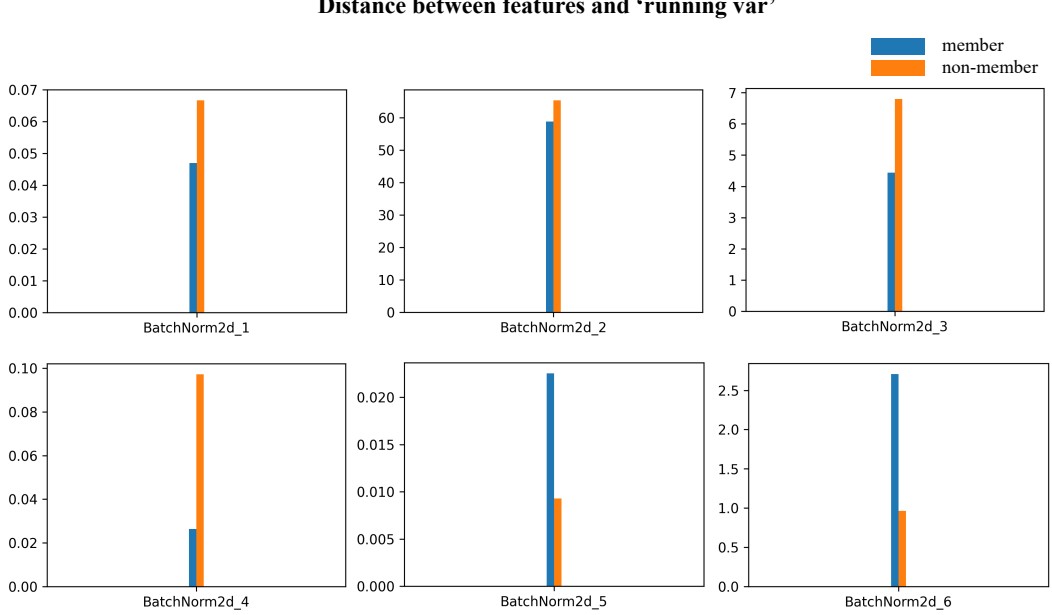

Figure 2: Visualization of distances between the intermediate features and "running var" for the Member and Non-member.

Table 1: The attack success rate of the membership inference attack in the case 1. We split the one dataset $\mathcal{D}$ (CASIA-WebFace) by half into $\mathcal{D}^{member}$ and $\mathcal{D}^{non-member}$. We also consider the case where the target models are trained with randomly flipped images, denoted as 'FR(flip)'.

| $\mathcal{D}_p$ | Proportion | | | | | |
|---|---|---|---|---|---|---|
| | 1% | 5% | 10% | 1%+FR(flip) | 5%+FR(flip) | 10%+FR(flip) |
| ASSD [1] | 57.89 | 59.76 | 59.63 | 59.73 | 58.95 | 62.04 |
| $\mathcal{A}_{mean}$ | 74.66 | 75.96 | 76.01 | 71.82 | 73.00 | 73.29 |
| $\mathcal{A}_{var}$ | 59.05 | 60.60 | 60.97 | 57.78 | 61.31 | 62.47 |
| $\mathcal{A}_{mean\&var}$ | 75.28 | 75.99 | 76.60 | 71.98 | 72.74 | 73.21 |
| $\mathcal{A}_{mean\&flip}$ | **91.23** | 92.14 | 92.82 | **87.22** | **86.81** | **87.33** |
| $\mathcal{A}_{mean\&var\&flip}$ | 90.91 | **92.28** | **93.22** | 83.31 | 86.09 | 87.20 |

Figure 3: More results of the model inversion attack in case 2.