# OpenReview forum: "Inference Attacks Against Face Recognition Model without Classification Layers"
_NeurIPS.cc/2023/Conference — Submitted to NeurIPS 2023_

### Official Review · Reviewer_wFe5 · 2023-07-05

**Soundness:** 3 good
**Presentation:** 3 good
**Contribution:** 2 fair
**Rating:** 5
**Confidence:** 4

**Summary:**

This paper introduces a novel inference attack algorithm for face recognition models that do not have classification layers. The proposed attack consists of two stages: membership inference and model inversion. The membership inference attack analyzes the distances between intermediate features and batch normalization parameters to determine if a face image belongs to the training dataset. The model inversion attack reconstructs sensitive private data using a pre-trained generative adversarial network (GAN) guided by the attack model.

**Strengths:**

1) The paper introduces a novel two-stage attack algorithm for face recognition models without classification layers.
2) The proposed method outperforms state-of-the-art similar works and can recover the identities of some training members.
3) This research has implications for the development of more robust and privacy-preserving face recognition models.

**Weaknesses:**

1) This paper would be beneficial to compare the proposed method with more state-of-the-art techniques to demonstrate its superiority.
2) This paper does not provide any code implementation.
3) The model performance used in the experiment is relatively low, and we hope to conduct experiments with models at a higher level of accuracy. (For example, ResNet200/VIT-Large on WebFace260M).

**Questions:**

Please refer to the weaknesses section.

**Limitations:**

Please refer to the weaknesses section.

---

> ### Author Rebuttal · Authors · 2023-08-09
>
> Thanks very much for constructive comments.
>
> 1、To our knowledge, there is currently no research on model inversion attacks against face recognition models without classification layers, i.e., there is no method to generate any training images in such a scenario. We are the first to explore this problem, and our work takes a step closer to its solution. Our attack scenario is first proposed for face recognition tasks, and there are no related methods for comparison.
>
> 2、We provide our codes in an anonymous repo：https://anonymous.4open.science/r/Attack_no_head-6470/ .
>
> 3、WebFace260M is a huge dataset, and we are unable to download it and re-train the model within a short time frame. As an alternative, we trained IR-SE-200 on MS1M-ArcFace dataset as the target model and present the corresponding experimental results in the material (Tab. 1).

---

> ### Author Response · Authors · 2023-08-16
>
> Dear Reviewer,
>
> We thank you very much for the precious review time and valuable comments. We have provided responses to your question and the weakness you mentioned. We hope this can address your concerns.
>
> We sincerely hope to further discuss with you whether or not your concerns have been addressed appropriately. Could you please let us know if you have additional questions or comments? We look forward to hearing from you soon. ：）
>
> Best regards,
>
> Authors

---

> > ### Author Response · Authors · 2023-08-20
> >
> > We have conducted the experiment you required, where we train our inference membership attack model with 50k training samples. And we provide the results as follows:
> >
> >
> > \begin{matrix}
> > 	Backbone(Dataset) & Case 1& Case 2 \\\\
> > 	  ResNet200(WebFace260M)&  95.38& 91.86 \\\\
> > \end{matrix}
> > We are looking forward to your reply. Thank you for your efforts in this paper.

---

> > > ### Comment · Reviewer_wFe5 · 2023-08-21
> > >
> > > Thank you for author's response. I also considered the feedbacks of other reviewers and adjusted my rating accordingly.

---

> > > > ### Author Response · Authors · 2023-08-21
> > > >
> > > > We deeply appreciate your recognition of our work :)

---

### Official Review · Reviewer_Ke6N · 2023-07-05

**Soundness:** 4 excellent
**Presentation:** 3 good
**Contribution:** 3 good
**Rating:** 8
**Confidence:** 5

**Summary:**

The authors propose a membership inference attack against face recognition (FR) models in the white-box scenario where membership information is known for some records and white-box model access is available, but without access to a classification layer. The attack utilizes information stored in batch-norm statistics and using a meta-classifier, the authors demonstrate the effectiveness of the proposed attack. They also extend the attack to improve model inversion attacks by utilizing their membership-classifier to "reject" generated samples that fall below a certain threshold.

**Strengths:**

- The utilization of an attack meant for one kind of privacy leakage (MI) in another scenario (model inversion) is interesting.

- The paper is well written, and proposed techniques/motivation are explained properly.

**Weaknesses:**

- L16: "..very first....without a classification layer." This is not the first work to explore FR models that do not use a classification layers. Please refer to [1, 2]. Similar claims appear on L39 about necessarily requiring logit access for good model performance. The authors in [1] report near-perfect detection for 3 different kinds of learning algorithms/models, none of which require classification logits.

- L25: "attribute attacks (also known as property inference attacks)" - these two are not the same at all. Similarly, L31-32 claim that all inference attacks on ML can be categorized into membership and model inversion attacks. Please refer to [3] for a detailed explanation and to better understand these differences.

- L188: "If it is from the training data set, then the attack model should output 1, otherwise we expect it to output 0" -how is this membership information obtained? As also outlined in Algorithm 1, the attack very clearly assumes access to not only batch-normalization parameters (which can only realistically come from full-model white-box access), but also knowledge of $m$ members and $n$ non-members. While the latter is reasonable, assuming knowledge of members is a very strong assumption (on top of an already-strong access model).

- L292 says "...theoretically analyze.." but nowhere in the paper did I see any theoretical analysis?

# Minor comments
- Figure 2: Why is Stage 2 on the left? It seems counter-intuitive.

## References

[1] Chen, Min, et al. "FACE-AUDITOR: Data Auditing in Facial Recognition Systems." USENIX, 2023

[2] Li, G., S. Rezaei, and X. Liu. "User-Level Membership Inference Attack against Metric Embedding Learning." ICLR 2022 Workshop on PAIR2Struct 2022.

[3] Salem, Ahmed, et al. "SoK: Let the Privacy Games Begin! A Unified Treatment of Data Inference Privacy in Machine Learning." 2023 IEEE S&P, 2023.

**Questions:**

- L100: There is no explicit mention of Differential Privacy, which is surprising. Are the authors implying that their attack can somehow break the guarantees provided by algorithms like DP-SGD?

- L244: Even 1% of 494K (CASIA) or 5.8M (MS1M), which would be ~5K or ~50K images respectively, is a **lot** of images. It would be more useful to have numbers for far lower percentages- perhaps 0.1% and 0.01%. Percentage is anyway not a useful indicator in a setting where there are millions of images. 0.1% may sound very low but is still more than 5K images for MS1M.

- Eq2. uses $\bar{v}_i - u_i$, but isn't $v_i$ itself $\frac{x - u_i}{\sigma_i}$? From my understanding of batch-norm (and what is also the case in most implementation),  batch-norm layers default to also using a running variance estimate in addition to the mean estimates.

**Limitations:**

- Section 3.1 - most FR models are not trained in the way that the authors describe (and assume in their experiments) here. The norm is to focus on training good embedding models so that models can be scaled easily to enroll future participants. As expected, training the model repeatedly whenever a new user ("class") is added is not optimal.

- Figure 3: Model inversion is supposed to recover actual records from the training data, but looking at the images in the figure that doesn't seem to be the case. Many faces (like second from left, second from right in first 2 rows) are not the same people, but rather people that "look like" each other.

---

> ### Author Rebuttal · Authors · 2023-08-09
>
> Thanks very much for careful and detailed comments.
> 1. In your reference [1],  they used face embeddings for the attack, but they did not achieve model inversion attacks. (Coincidentally, this article was published in April, whereas our work was also completed in April.) In [2], we mentioned it and compared our work with it in L. 102 of the paper. Face recognition tasks are more complex than pedestrian re-identification tasks (as you mentioned, the training dataset is larger and has more categories). Moreover, these works do not include any model inversion attack, while our work is the first to implement a model inversion attack algorithm in this scenario.
>
> 2. L.25 is a mistake, and we apologize for it. We are aware that there are other types of inference attacks, and we wrote this sentence based on reference [1]. We will correct it according to the literature you provided.
>
> 3. In Section 4, we have discussed experimental details. Firstly, we consider two common different assumptions: the one case where the attackers have access to partial training data and the other case where the attackers have no access to any data (a common setting for most attack methods). In the former case, we can directly obtain membership information. In the latter case, we will use the shadow model method followed by [2], and more details are described in the paper L. 246.
>
> 4. We did not give theoretical analysis, and our method is a heuristic algorithm like [3][4].
>
> 5. DP-SGD is a defense method that is beyond the scope of our study. We only focus on researching model inversion attacks, not evaluating defense methods. We only discuss about attack algorithms as [3][4][5].
>
> 6. The proportion of training data used for training attack models in [4][5] is also very high (2000 out of 16522). Moreover, for face datasets, obtaining 5k face images from public datasets is not difficult, and our method is also applicable for training attack models (using shadow models) on publicly available datasets. Additionally, we have supplemented the attack experimental results with fewer face images in the material (Tab. 1).
>
> 7. A similar approach was adopted in [4][5], where they compared the distance between sample features and reference sets to determine if a sample belongs to the training data. Additionally, we also provide the experimental results of the method you provided in the material (Tab. 1).
>
> 8. Face models are usually trained following the paradigm we mentioned in the paper (refer to [6,7]). Our FR model is also designed to train a good FR embedding representation model,  and the norm is to focus on training good embedding models so that models can be scaled easily to enroll future participants, so increasing the number of participants  needn't retrain the model.
>
> 9. To our knowledge, there is currently no research on model inversion attacks against face recognition without classification layers, i.e., there is no method to generate any training images in such a scenario. We are the first to explore this problem, and our work takes a step closer to its solution. Our attack scenario is first proposed for face recognition tasks, and there are no related methods for comparison.
>
>
>
> [1]Comprehensive Privacy Analysis of Deep Learning: Passive and Active White-box Inference Attacks against Centralized and Federated Learning.
>
> [2]Liu, Yugeng, et al. "{ML-Doctor}: Holistic Risk Assessment of Inference Attacks Against Machine Learning Models." 31st USENIX Security Symposium (USENIX Security 22). 2022.
>
> [3]Plug & play attacks: Towards robust and flexible model inversion attacks. In Proceedings of the 39th International Conference on Machine Learning (ICML), Proceedings of Machine Learning Research, pages 20522–20545. PMLR, 2022.
>
> [4]Li, G., S. Rezaei, and X. Liu. "User-Level Membership Inference Attack against Metric Embedding Learning." ICLR 2022 Workshop on PAIR2Struct 2022.
>
> [5]Gao, Junyao, et al. "Similarity Distribution based Membership Inference Attack on Person Re-identification." Proceedings of the AAAI Conference on Artificial Intelligence. Vol. 37. No. 12. 2023.
>
> [6]"Arcface: Additive angular margin loss for deep face recognition." Proceedings of the IEEE/CVF conference on computer vision and pattern recognition. 2019.
>
> [7]Kim, Minchul, Anil K. Jain, and Xiaoming Liu. "Adaface: Quality adaptive margin for face recognition." Proceedings of the IEEE/CVF conference on computer vision and pattern recognition. 2022.

---

> > ### Comment · Reviewer_Ke6N · 2023-08-10
> > **Most issues resolved**
> >
> > Thanks for responding to my concerns. I will raise my rating to reflect the same :) While most of my concerns seem to have been resolved, I do have one main issue:
> >
> > The proposed work indeed related to model inversion, but uses membership inference from the first stage and is thus dependent on the performance of the first stage. While DP-SGD (or other heuristic methods, like noise in predictions) in general would not help defend against model inversion, it would make the first stage's performance drop significantly, which in turn would lead to much lower model inversion performance. This is why I am curious to see how a defense against MI would affect the overall performance of the proposed inversion attack, since it ultimately utilizes membership inference.
> >
> > Some other minor comments:
> >
> > - Since the proposed algorithm is indeed heuristic, I would urge the authors to remove "theoretically analyze" on L292 since it borders on being misleading.
> >
> > - My question regarding running variance remains - could you please furher elaborate on that?

---

> > > ### Author Response · Authors · 2023-08-10
> > >
> > > Thank you very much for providing valuable suggestions.
> > >
> > > The attack we performed is against FR models without a classification layer. If DP-SGD is added to a model with a classification layer, it may affect the parameters of the classification layer besides that of the backbone. Changes in the parameters of the classification layer may lead to a decrease in the effectiveness of the attack. Our attack method does not use the classification layer. We speculate that in our scenario, for the same privacy budget, our attack method may not be very sensitive to the perturbations of DP-SGD. Therefore, the impact of DP-SGD on membership inference attacks may not be as significant as the former. This also will be the focus of our future research.
> > >
> > > We will delete the relevant parts according to your opinion.
> > >
> > > We have addressed your concerns about running variance and supplemented the experimental results of the ideas you proposed in 7. The mean is the first-order moment statistic and the variance is the second-order moment statistic. We speculate that the reason for the observation in 7 may be that the feature space is the Euclidean space, and the feature distance measurement we used may be more compatible with the first-order moment, thereby performing better than using the second-order moment.

---

> > > > ### Comment · Reviewer_Ke6N · 2023-08-10
> > > >
> > > > If running an experiment on DP-SGD trained models is not too time consuming, it would be better to have experimental results to back the claim made above. If training a model with DP noise would be too time-consuming, I would urge the authors to at least experiment with some heuristic defense, like [this](https://arxiv.org/pdf/2307.01610.pdf)

---

> > > > > ### Author Response · Authors · 2023-08-16
> > > > >
> > > > > Thank you very much for your positive feedback. We use the naive DP-SGD method to protect the face recognition model referring to [1]. Meanwhile, we adopt a heuristic attack algorithm [2] which uses the output of the model's classification layer as input of the attack model, serving as the baseline. For different privacy budget settings, we conduct attack experiments and demonstrate the results as follows.
> > > > >
> > > > > $$\begin{matrix}
> > > > > \epsilon&1.031|&3.413|&5.122|&10.243\\\\
> > > > > FR_{acc}&62.29|&92.53|&93.24|&93.31\\\\
> > > > > bsl(ASR)&50.00|&55.30|&56.35|&60.78\\\\
> > > > > ours(ASR)&50.00|&88.20|&90.67|&93.25
> > > > > \end{matrix}$$
> > > > >
> > > > > In the table, "FR_acc" denotes the average accuracy of the target FR model evaluating 6 benchmarks(LFW, CFP-FP, AgeDB, CPLFW,  CALFW and VGG2), "bsl" denotes the attack algorithm using the logit of the classification layer to perform member inference attack [2].
> > > > > We find that in larger $\epsilon$, our algorithm still maintains the ability for membership inference attack, and our algorithm's ASR (Attack Success Rate) is much higher than that of the baseline. However, when $\epsilon$ is lower, the accuracy of the face recognition model sharply decreases, and neither our algorithm nor the baseline can perform membership inference attack on the model.
> > > > >
> > > > > [1]Abadi, Martin, et al. "Deep learning with differential privacy." Proceedings of the 2016 ACM SIGSAC conference on computer and communications security. 2016.
> > > > >
> > > > > [2]Liu, Yugeng, et al. "{ML-Doctor}: Holistic Risk Assessment of Inference Attacks Against Machine Learning Models." 31st USENIX Security Symposium (USENIX Security 22). 2022.

---

> > > > > > ### Comment · Reviewer_Ke6N · 2023-08-16
> > > > > >
> > > > > > Thanks for running the experiments on such short notice! I think this takes care of my last concern, and should serve as strong evidence for the utility of the attack.
> > > > > >
> > > > > > Just to be clear, when you say you used [2] as a baseline, do you run their attack for model inversion, or use their MI attack as a swap-in for your own MI attack and then continue to run your own model inversion?
> > > > > >
> > > > > > A few suggestions:
> > > > > >
> > > > > > - Although the trend with $\epsilon$ values makes it clear that the proposed method is indeed better, this is a very unusual selection of $\epsilon$ values. I would recommend trying $1, 2, 5, 10$. I would've suggested $0.1$ and lower, but looks like at even $1$ ASR is equivalent to random guessing.
> > > > > >
> > > > > > - When reporting in the paper, I would recommend having dataset-wise numbers in the Appendix for completeness. Also, if space permits, do include task-accuracies for these $\epsilon$ values. For readers not familiar with DP, it would help put in perspective the cost at which such a technique can provide protection (since task accuracy takes a big hit with low $\epsilon$ values in DP).

---

> > > > > > > ### Author Response · Authors · 2023-08-16
> > > > > > >
> > > > > > > Thanks again for giving pertinent and detailed suggestions. First, we use their method [2] to perform a membership inference attack and do not perform a model inversion attack further. Our table shows the success rate of member inference attacks.  And, according to previous experience, a good member inference attack model will contribute to a good model inversion attack result. In addition, after careful communication with you, we have done some in-depth experiments and got surprising and strong results. We will add them to the paper or appendix as you suggested.
> > > > > > >
> > > > > > > Unfortunately, so far, except for you,  other reviewers do not discuss this with us in significant depth. I'm afraid it's difficult to move to the next stage with the current score. Of course, we also look forward to the opportunity to communicate with other reviewers. Thank you again. : )

---

> > > > > > > > ### Comment · Reviewer_Ke6N · 2023-08-16
> > > > > > > >
> > > > > > > > I am glad to hear that my feedback helped the authors improve their paper :) I'm hoping that the Area Chair will nudge authors directly to be more active in participating. Most of the reviewers seem to have provided very primitive and vague feedback. I think this is a good paper, and have increased my score to reflect this belief.
> > > > > > > >
> > > > > > > > These two are suggestions for a later stage, not a requirement. If and when the authors get the time, it would be useful to think about these:
> > > > > > > >
> > > > > > > > > ...we use their method [2] to perform a membership inference attack...
> > > > > > > >
> > > > > > > > I see. In that case, I would suggest sticking to more recent and state-of-the-art attacks in a camera-ready version, like [1] or [2].
> > > > > > > >
> > > > > > > > > ...according to previous experience, a good member inference attack model will contribute to a good model inversion attack result.
> > > > > > > >
> > > > > > > > True, but for completeness results should be included for an end to end model inversion attack as well. Although MI is used as a swap-in module, the kind of data that different attacks are good/bad at for inferring membership may impact downstream model inversion in different ways.
> > > > > > > >
> > > > > > > > ## References
> > > > > > > >
> > > > > > > > [1] Ye, Jiayuan, et al. "Enhanced membership inference attacks against machine learning models." Proceedings of the 2022 ACM SIGSAC Conference on Computer and Communications Security. 2022.
> > > > > > > >
> > > > > > > > [2] Carlini, Nicholas, et al. "Membership inference attacks from first principles." 2022 IEEE Symposium on Security and Privacy (SP). IEEE, 2022.

---

> > > > > > > > > ### Author Response · Authors · 2023-08-20
> > > > > > > > >
> > > > > > > > > We will follow the suggestions you have provided and supplement above attack experiments for completeness :)

---

### Official Review · Reviewer_X6ex · 2023-07-07

**Soundness:** 2 fair
**Presentation:** 2 fair
**Contribution:** 2 fair
**Rating:** 4
**Confidence:** 3

**Summary:**

The paper presents a novel method for inference attacks against face recognition method. In particular, it advocates two-stage inference attack, where the first stage identify the membership and the second stage involves model inversion attack that recover the input from embedding. Experimental evaluation shows that the proposed method can largely identify the correct membership and the model inversion sees good as well.

**Strengths:**

1. It proposes a novel two-stage method for inference attack of face recognition systems.

2. Experimental evaluation shows the proposed method has promising results.

**Weaknesses:**

1. The evaluation is not thorough. It only conduct some ablation study with comparing to the existing attack methods.

2. The paper does not cover some import topic for inference attack of face recognition, such as the black box attacks for face recognition.

**Questions:**

1. Recovering input images from embedding has been widely studied and employed. What are the difference between the proposed method with the exiting decoding methods?

**Limitations:**

unavailable.

---

> ### Author Rebuttal · Authors · 2023-08-09
>
> Thanks very much for your insightful reviews.
> 1. As far as we know, there is no standard evaluation for model inversion attacks without classification layers. Our work is the first to focus on model inversion attacks under such a scenario, i.e., there is no method to generate any training images in such a scenario. We are the first to explore this problem, and our work takes a step closer to its solution.
> 2. We focus on white-box model inversion attacks that are closer to practical applications, i.e., without classification layers. For example, [1] also only considers white-box model inversion attacks against FR models, but with classification layers.
>
> 3. It is possible to reconstruct the input image using GAN or VAE models by the embedding, which is obtained by the model. However, our task is model inversion attack, which means recovering the training data as much as possible from the pre-trained model. We first generate a batch of candidate images using GAN, and then obtain the embeddings of these candidate images using the FR model. Besides, we train an attack model to obtain the embedding distribution of the real training data. Finally, we optimize the embeddings of the candidate images to approach the distribution of the real training data embeddings, in order to recover the training data. The main difference is that, the former attempts to generate high-fidelity images from the embedding, while our goal is to recover training data.
>
> [1]Plug & play attacks: Towards robust and flexible model inversion attacks. In Proceedings of the 39th International Conference on Machine Learning (ICML), Proceedings of Machine Learning Research, pages 20522–20545. PMLR, 2022.

---

> ### Author Response · Authors · 2023-08-16
>
> Dear Reviewer,
>
> We thank you very much for the precious review time and valuable comments. We have provided responses to your question and the weakness you mentioned. We hope this can address your concerns.
>
> We sincerely hope to further discuss with you whether or not your concerns have been addressed appropriately. Could you please let us know if you have additional questions or comments? We look forward to hearing from you soon. ：）
>
> Best regards,
>
> Authors

---

> > ### Author Response · Authors · 2023-08-21
> >
> > Dear Reviewer,
> >
> > Thanks for your careful comments.
> >
> > For your Weaknesses 1, we have done some additional experiments to demonstrate the effectiveness of our proposed approach.
> >
> > Firstly, we supply some experiments against DP-SGD [1] serving as strong evidence for the utility of the attack. Meanwhile, we adopt a heuristic attack algorithm [2] which uses the output of the model's classification layer as input of the attack model, serving as the baseline. For different privacy budget settings, we conduct attack experiments and demonstrate the results as follows.
> > $$\begin{matrix}
> > \epsilon &1.031&3.413 &5.122 &10.243 \\\\
> > FR_{acc} &62.29&92.53&93.24&93.31\\\\
> > bsl(ASR) &50.00&55.30&56.35&60.78\\\\
> > ours(ASR)&50.00&88.20&90.67&93.25\\\\
> > \end{matrix}$$
> > In the table, "FR_acc" denotes the average accuracy of the target FR model evaluating 6 benchmarks(LFW, CFP-FP, AgeDB, CPLFW, CALFW and VGG2), "bsl" denotes the attack algorithm using the logit of the classification layer to perform member inference attack [2]. We find that in larger $\epsilon$
> > , our algorithm still maintains the ability for membership inference attack, and our algorithm's ASR (Attack Success Rate) is much higher than that of the baseline. However, when $\epsilon$
> >  is lower, at the cost of the accuracy of FR model sharply decreasing, neither our algorithm nor the baseline can perform membership inference attack on the model.
> >
> > Secondly, we also perform the attack experiment on the model at a higher level of accuracy (i.e. ResNet200 training on WebFace260M) to demonstrate our method effectiveness, where we train our inference membership attack model with 50k training samples. And we provide the results as follows:
> > $$\begin{matrix}
> > Backbone(Dataset) & Case 1&Case 2 \\\\
> > ResNet200(WebFace260M)&95.38&91.86\\\\
> > \end{matrix}$$
> >
> > For your Weakness 2, considering BN is the internal parameter in the model, we try replacing BN with the output embedding to perform black-box attacks for face recognition. We provide some experiment results of the first stage (i.e. member inference attack) as follows:
> >
> > $$\begin{matrix}
> > Backbone(Dataset) & IR-SE-50 (CASIA)&IR-SE-50 (MS1M-ArcFace) \\\\
> > ASR &63.24&63.94\\\\
> > \end{matrix}$$
> >
> > The attack success rate (ASR) is much lower than our proposed methods. Therefore, we do not suppose such a member inference attack model would contribute to a good model inversion attack result. In this way, the model inversion attack results heavily rely on the ability of the pre-trained GAN. As such, this scenario falls outside the scope of our research.
> >
> > We are looking forward to your reply and thanks for your efforts in this paper.
> >
> > [1]Abadi, Martin, et al. "Deep learning with differential privacy." Proceedings of the 2016 ACM SIGSAC conference on computer and communications security. 2016.
> >
> > [2]Liu, Yugeng, et al. "{ML-Doctor}: Holistic Risk Assessment of Inference Attacks Against Machine Learning Models." 31st USENIX Security Symposium (USENIX Security 22). 2022.

---

### Official Review · Reviewer_LWa5 · 2023-07-07

**Soundness:** 3 good
**Presentation:** 3 good
**Contribution:** 2 fair
**Rating:** 3
**Confidence:** 4

**Summary:**

In this submission, the authors advocate an inference attack composed of two stages for practical FR models. The first stage analyzes the distances between the intermediate features and batch normalization parameters. The second stage reconstructs data using a pre-trained generative adversarial network (GAN) guided by the attack model in the first stage.

**Strengths:**

1.The writing and presentation are good and easy to follow.

2.The experimental results also demonstrate some effectiveness of the proposed method.

**Weaknesses:**

1.The overall technical contributions are somewhat limited, firstly, the way of using BN to perform membership inference attack has been explored for a long time. And secondly, the inversed training data are from a pretrain GAN, which is heavily depending on the strength of the pretrain GAN. And optimizing the synthesized face data is too weak only by the single supervision from the first stage.

2.And from the Figure 3, I don’t think the recovered face data is visually close to the original data for some of the pairs. Therefore, I doubt that whether the authors really achieve the initial goal, recovering the similar enough or effective enough face training data, by their proposed method or not.

3.The experimental comparisons are too simple and rough, lacking some important state-of-art related competitors.

**Questions:**

Please see the details in Weaknesses.

**Limitations:**

This submission has adequately addressed the limitations.

---

> ### Author Rebuttal · Authors · 2023-08-09
>
> Thanks very much for constructive and insightful suggestions.
> 1. It is true that the way of using BN to perform membership inference attacks has been explored, but this paper focuses on the model inversion attack in the field of face recognition without classification layers, which is the very first exploration in such a scenario as far as we know. GAN is currently a commonly used method for face reconstruction attacks. The reviewer can refer to the references [1][2][3]. The supervision we proposed plays an important role in the model inversion attack. We provide some results which are generated without our supervision in the material (Fig. 1). We would be grateful if you could provide more detailed advice and instructions.
> 2. To our knowledge, there is currently no research on model inversion attacks against face recognition without classification layers, i.e., there is no method to generate any training images in such a scenario. We are the first to explore this problem, and our work takes a step closer to its solution. Our attack scenario is first proposed for face recognition tasks, and there are no related methods for comparison.
>
>
> [1]Mirror: Model inversion for deep learning network with high fidelity. In Proceedings of the 29th Network and Distributed System Security Symposium.
>
> [2]Plug & play attacks: Towards robust and flexible model inversion attacks. In Proceedings of the 39th International Conference on Machine Learning (ICML), Proceedings of Machine Learning Research, pages 20522–20545. PMLR, 2022.
>
> [3]Model inversion attack by integration of deep generative models: Privacy-sensitive face generation from a face recognition system. IEEE Transactions on Information Forensics and Security

---

> ### Author Response · Authors · 2023-08-16
>
> Dear Reviewer,
>
> We thank you very much for the precious review time and valuable comments. We have provided responses to your question and the weakness you mentioned. We hope this can address your concerns.
>
> We sincerely hope to further discuss with you whether or not your concerns have been addressed appropriately. Could you please let us know if you have additional questions or comments? We look forward to hearing from you soon. ：）
>
> Best regards,
>
> Authors

---

> > ### Author Response · Authors · 2023-08-21
> >
> > Dear Reviewer,
> >
> > Thanks very much for your constructive and insightful suggestions.
> > For your Weaknesses 1 and 3, we have done some additional experiments to demonstrate the effectiveness of our proposed approach.
> >
> > For Weakness 1, our work is indeed related to model inversion, but uses membership inference from the first stage and is thus dependent on the performance of the first stage. Besides, we generated some results without supervision from the first stage using a pre-trained GAN (shown in our Rebuttal PDF). We find that with supervision from the first stage, the generated results are even further closer to the target IDs.
> >
> > For Weakness 3, we supply some experiments against DP-SGD [1] serving as strong evidence for the utility of the attack. Meanwhile, we adopt a heuristic attack algorithm [2] which uses the output of the model's classification layer as input of the attack model, serving as the baseline. For different privacy budget settings, we conduct attack experiments and demonstrate the results as follows.
> >
> > $$\begin{matrix}
> > \epsilon &1.031&3.413 &5.122 &10.243 \\\\
> > FR_{acc} &62.29&92.53&93.24&93.31\\\\
> > bsl(ASR) &50.00&55.30&56.35&60.78\\\\
> > ours(ASR)&50.00&88.20&90.67&93.25\\\\
> > \end{matrix}$$
> > In the table, "FR_acc" denotes the average accuracy of the target FR model evaluating 6 benchmarks(LFW, CFP-FP, AgeDB, CPLFW, CALFW and VGG2), "bsl" denotes the attack algorithm using the logit of the classification layer to perform member inference attack [2]. We find that in larger $\epsilon$
> > , our algorithm still maintains the ability for membership inference attack, and our algorithm's ASR (Attack Success Rate) is much higher than that of the baseline. However, when $\epsilon$
> >  is lower, at the cost of the accuracy of FR model sharply decreasing, neither our algorithm nor the baseline can perform membership inference attack on the model.
> >
> > For Weakness 3, we  also perform the attack experiment on the model at a higher level of accuracy (i.e. ResNet200 training on WebFace260M) to demonstrate our method effectiveness, where we train our inference membership attack model with 50k training samples. And we provide the results as follows:
> >
> > $$\begin{matrix}
> > Backbone(Dataset) & Case 1&Case 2 \\\\
> > ResNet200(WebFace260M)&95.38&91.86\\\\
> > \end{matrix}$$
> >
> > For your Weakness 2, we acknowledge that the results of our attack are not looking as good as those achieved by other methods with classification layers. But to our knowledge, there is currently no research on model inversion attacks against face recognition without classification layers, i.e., there is no method to generate any training images in such a scenario. We are the first to explore this problem, and our work takes a step closer to its solution.
> >
> > We are looking forward to your reply. Thank you for your efforts in this paper.
> >
> > [1]Abadi, Martin, et al. "Deep learning with differential privacy." Proceedings of the 2016 ACM SIGSAC conference on computer and communications security. 2016.
> >
> > [2]Liu, Yugeng, et al. "{ML-Doctor}: Holistic Risk Assessment of Inference Attacks Against Machine Learning Models." 31st USENIX Security Symposium (USENIX Security 22). 2022.

---

### Author Rebuttal · Authors · 2023-08-09

As far as we know, model inversion attack against face recognition models without classification layers is a challenging task and is first proposed in this paper. This is a more practical scenario and there is currently no method to recover any training images in the scenario. We are the first to explore this problem, and our work takes a step closer to its solution. Our attack scenario is first proposed for face recognition tasks, and there are no related methods for comparison.

We supplement some experiment results in the material to address the reviewers' concerns. And we provide ur codes in an anonymous repo：https://anonymous.4open.science/r/Attack_no_head-6470/ .

---

> ### Author Response · Authors · 2023-08-22
>
> The attack scenario where face recognition models are without classification layers is closer to reality. Because the current large-scale applications of face recognition models are all without classification layer, such as face payment, ticketing systems, and so on. Our work is a review of the privacy leakage problem in large-scale real scenarios.

---

### Decision · Program_Chairs · 2023-09-21

**Decision:**

Reject

**Comment:**

The paper is reviewed by 4 reviewers. After author rebuttal and discussion period, final ratings are 3, 4, 5, and 8, an average rating of 5.

To begin with, the paper tackles membership inference and model inversion attacks for face recognition models without classification layer, noting that it is a more practical setting as FR models are typically deployed without classification layers. Membership inference attack is done by exploiting the BN stats to train a classifier using member and non-member training data. Model inversion attack makes use of StyleGAN to render a face image used in training, while harnessing the membership classifier trained in the first stage.

Reviewers raised concerns on 1) limited technical contribution (**LWa5**), 2) lack of experimental results (**LWa5**, **X6ex**, **wFe5**). For example, while the absence of classification layer for FR models necessitates alternatives, the use of BN stats for membership inference attack is not new, as pointed by **LWa5**. Experiments on model inversion attack is incomplete -- as pointed by reviewers (**LWa5**, **X6ex**), Figure 3 is a qualitative results without a metric on the similarity between original and inversion or a comparison. AC agrees that the experimental validation is too weak in the current form. Reviewer Ke6N strongly argued for acceptance of the paper with the addition of an analysis of DP. While it could be a good addition, but not essential, which is also mentioned by the authors as DP-SGD is beyond the scope of the work. The paper should be strengthened with more systematic analysis on the model inversion attack, both in terms of evaluation metric and comparison against others or with different design choices.

While the paper has studied a practically important problem and presented sound method, experimental validation should be strengthened. AC recommend for rejection.